

# Concentration-dependent polymorphism of insulin amyloid fibrils

Andrius Sakalauskas, Mantas Ziaunys and Vytautas Smirnovas

Institute of Biotechnology, Life Sciences Center, Vilnius University, Vilnius, Lithuania

## ABSTRACT

Protein aggregation into highly structured fibrils has long been associated with several neurodegenerative disorders, such as Alzheimer's or Parkinson's disease. Polymorphism of amyloid fibrils increases the complexity of disease mechanisms and may be one of the reasons for the slow progress in drug research. Here we report protein concentration as another factor leading to polymorphism of insulin amyloid fibrils. Moreover, our data suggests that insulin amyloid conformation can self-replicate only via elongation, while seed-induced nucleation will lead to environment-defined conformation of fibrils. As similar observations were already described for a couple of other amyloid proteins, we suggest it to be a generic mechanism for self-replication of different amyloid fibril conformations.

# INTRODUCTION

Protein aggregation into amyloid fibrils has been linked to multiple neurodegenerative disorders, including Alzheimer's, Parkinson's and infectious prion diseases (*Chiti & Dobson, 2017*; *Knowles, Vendruscolo & Dobson, 2014*), which affect tens of millions of people worldwide and is predicted to become even more prominent as the average human lifespan continues to increase (*Isik, 2010*). Matters are further complicated by the fact that very few drugs have reached stage four of clinical trials and no efficient treatment or cure is available (*Cummings et al., 2019*; *Mehta et al., 2017*). One of the main reasons for such limited progress in the development of potential cures may be the complexity of fibril formation mechanisms (*Meisl et al., 2017*; *Doig et al., 2017*; *Nasica-Labouze et al., 2015*), as well as polymorphism of amyloid aggregates (*Stein & True, 2014*).

The ability of the same protein to adopt distinct pathogenic conformations was first reported in studies of infectious prions and such conformations were referred to as strains (*Safar et al., 1998*; *Collinge & Clarke, 2007*). Recently strain-like polymorphism was reported for a number of amyloid proteins in vivo (*Lu et al., 2013*; *Watts et al., 2014*; *Fändrich et al., 2018*; *Yamasaki et al., 2019*), in vitro (*Heise et al., 2005*; *Paravastu et al., 2008*; *Debelouchina et al., 2010*; *Dinkel et al., 2011*; *Bousset et al., 2013*), and in silico (*Pellarin et al., 2010*). A number of environmental factors including pH (*Sneideris et al., 2015*), temperature (*Tanaka et al., 2006*; *Colby et al., 2009*), concentration of co-solvents (*Dzwolak et al., 2004*; *Chatani et al., 2012*), denaturants (*Colby et al., 2009*; *Cobb et al., 2014*) or salts (*Morel et al., 2010*; *Bousset et al., 2013*), as well as agitation (*Petkova et al.,*

Corresponding author
Vytautas Smirnovas,
vytautas@smirnovas.info

*2005*; *Ostapchenko et al., 2010*) can lead to different conformations of amyloid fibrils. Enormous amounts of data must be collected and analyzed in order to understand the complex effects of the environment on polymorphism of amyloids.

Due to its relatively low cost, wide availability and simple aggregation protocols, insulin became one of the most common proteins used to study amyloid fibril formation. Several years ago, we summarized available data on polymorphism of insulin amyloid fibrils and came with the hypothesis that the number of insulin amyloid conformations may be limited to two and the major factor which determines formation of different strains is a shift of the equilibrium between insulin monomers and dimers (oligomers) (*Sneideris et al., 2015*). Our current data supports the existence of a third conformation of insulin amyloid fibrils and suggests that polymorphism of insulin amyloid fibrils is more complex.

## MATERIALS AND METHODS

### Insulin sample preparation

Human recombinant insulin powder (Sigma-Aldrich cat. No. 91077C) was dissolved in a 20% acetic acid solution containing 100 mM NaCl (reaction solution) to a final concentration of 2 mM (11.6 mg/ml). Insulin concentration was determined by measuring the sample's absorbance at 280 nm $\varepsilon = 6,335\,\mathrm{M}^{-1}\mathrm{cm}^{-1}$, $M = 5808$ Da. Samples for unseeded aggregation kinetic measurements were prepared by diluting the 2 mM stock solution using the reaction solution and 10 mM ThT stock solution to a range of concentrations from 0.2 mM to 1.0 mM (which contained 100 µM of ThT). For seeded aggregation, insulin fibrils prepared from the 0.2 mM and 1.0 mM samples were sonicated for 10 min using Sonopuls 3100 (Bandelin) ultrasonic homogenizer equipped with a MS73 tip (40% power, 30 s sonication/ 30 s rest intervals). The homogenized fibrils were then diluted with the reaction solution and mixed with the 2 mM insulin and 10 mM ThT stock solutions to yield 0.2 mM and 1.0 mM concentration samples containing 100 µM ThT and a range of fibril concentrations (from 5% to $10^{-6}$ % of monomer mass).

### Aggregation kinetics

Insulin aggregation kinetics were monitored at 60 °C without agitation by measuring ThT fluorescence emission intensity (excitation wavelength—440 nm, emission—480 nm) through the bottom of a 96 well non-binding surface plate using Synergy H4 Hybrid Multi-Mode (Biotek) plate reader (readouts were taken every 10 min to minimize plate agitation). For every condition, four independent measurements were performed. Aggregation half-times ($t_{50}$) were calculated as the time needed to reach 50% of signal intensity. The full concentration range of aggregated insulin samples were centrifuged at 10 000 g for 30 min and the residual unaggregated insulin in the supernatant was determined to be less than 1% of initial protein concentration.

### Atomic Force Microscopy (AFM)

After kinetic measurements, samples were diluted with the reaction solution to a 50 µM concentration and 20 µL of each was deposited on freshly cleaved mica and incubated for 1 min. Subsequently, samples were rinsed with one mL of MilliQ water and dried under

gentle airflow. Three-dimensional AFM maps were acquired using a Dimension Icon (Bruker) atomic force microscope operating in tapping mode and equipped with a silicon cantilever Tap300AI-G (40 N m$^{-1}$; Budget Sensors) with a typical tip radius of curvature of 8 nm. High-resolution (1,024 × 1,024 pixels) images were acquired. The scan rate was 1 Hz. AFM images were flattened and analyzed using SPIP (Image Metrology).

## Fourier-Transform Infrared (FTIR) Spectroscopy

Insulin fibrils were separated from solution by centrifugation at 10 000 g for 30 min and subsequently resuspended in one mL of $D_2O$, the procedure was repeated three times. Then the fibrils were resuspended in 0.2 mL of $D_2O$ and sonicated for 1 min using a MS72 tip (with 20% power and constant sonication). Samples were deposited between two $CaF_2$ transmission windows separated by 0.05 mm teflon spacers. The FTIR spectra were recorded using Vertex 80v (Bruker) IR spectrometer equipped with a mercury cadmium telluride detector, at room temperature under vacuum (∼2 mBar) conditions. 256 interferograms of two cm$^{-1}$ resolution were averaged for each spectrum. Spectrum of $D_2O$ was subtracted from the spectrum of each sample. All spectra were normalized to the same area of amide I/I' band (1,700–1,595 cm$^{-1}$). All data processing was performed using GRAMS software.

## RESULTS

### Fibril formation at different concentrations

Aggregating a range of insulin concentrations in 20% acetic acid with 100 mM NaCl at 60 °C without agitation reveals a typical kinetic curve pattern, where an increasing insulin concentration leads to shorter aggregation times (Fig. 1A). However, we observe an uneven ratio distribution between ThT fluorescence emission intensities and final fibril concentrations (Fig. 1B). As the concentration of insulin in the sample increases, this ratio shifts ten-fold, indicating either a higher quantum yield or considerably more bound ThT molecules.

The FTIR spectra of aggregated samples exhibit maxima in the amide I/I' region at ∼1,628 cm$^{-1}$ with the shoulder at ∼1 641 cm$^{-1}$, and a small band outside of the amide I/I' region at ∼1,729 cm$^{-1}$ (Fig. 1C), which is very similar to previously reported insulin fibrils formed in phosphate buffer at pH ≤2 (*Sneideris et al., 2015*). However, minor concentration-dependent differences can be observed (Figs. 1C and 1D). The spectra of fibrils, formed at lower insulin concentrations, have a pronounced shoulder at 1,641 cm$^{-1}$, while a minor band at 1,620 cm$^{-1}$ appears in the second derivative spectra (Fig. 1D) of samples aggregated at higher protein concentrations. The 1.0 mM and 0.8 mM fibril spectra are nearly identical, while 0.6 mM and 0.4 mM spectra appear to be intermediates between 0.8 mM and 0.2 mM, suggesting the existence of two distinct conformations.

### Fibril morphology

The morphology of insulin fibrils formed at different concentrations was compared using AFM. We can see far more small and separated aggregates in samples formed at lower insulin concentration (Figs. 2A–2E). Analysis of variance (ANOVA) reveals that there is a statistically relevant fibril height difference ($p = 0.01$, $n = 50$) between the low and high

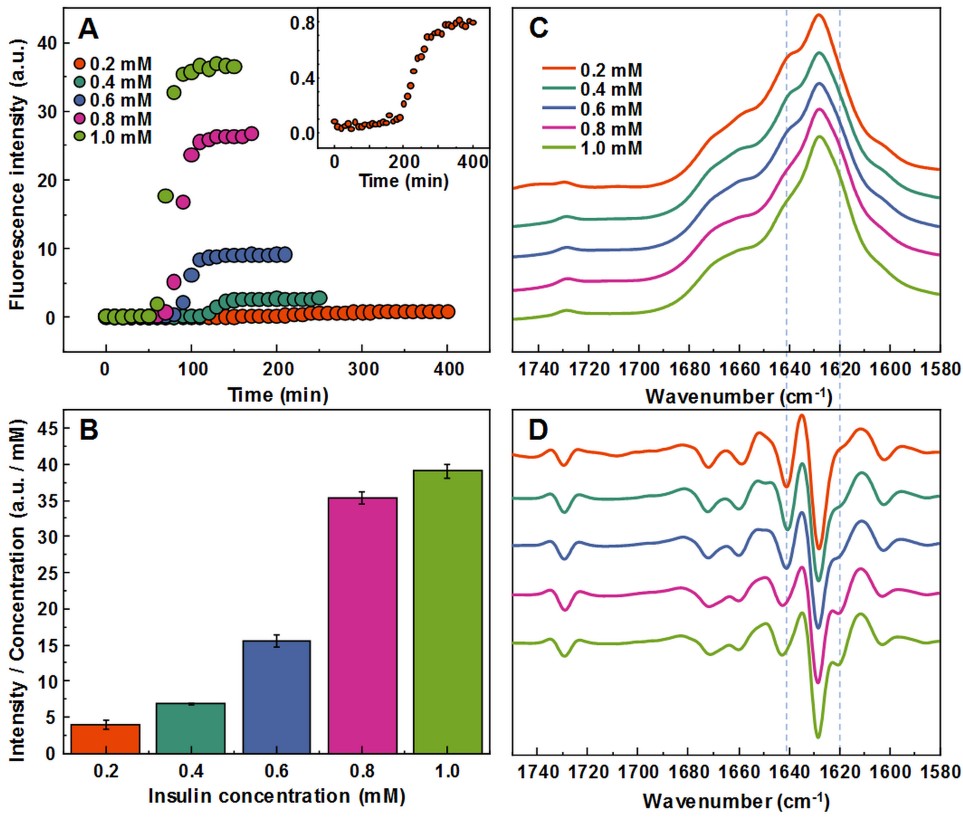

**Figure 1** **Concentration-dependent differences of insulin aggregation.** Aggregation kinetics of un-seeded insulin in 20% acetic acid with 100 mM NaCl at 60 °C without agitation followed by ThT fluores-cence (A), insert shows aggregation kinetics of 0.2 mM insulin. Each kinetic data point is the average of 4 repeats. ThT fluorescence intensity and fibril concentration ratios (B). FTIR absorption (C) and second derivative (D) spectra of insulin fibrils.

concentration samples (Fig. 2F). Additional AFM images of these conditions are available as Fig. S1.

## Seeded aggregation

In order to determine whether observed different fibril templates can propagate at unfavorable conditions, a set of seeded aggregation reactions were performed (Figs. 3A–3D, Fig. S2). In all four cases we observe a fibril-concentration-dependent seeding propensity (Fig. 3E), however, there is an interesting ThT fluorescence distribution, based on the amount and type of seed added (Fig. 3F). When the low concentration fibrils (0.2 mM, further on referred as LCF) are added to 0.2 mM insulin solutions, there are relatively no major differences in the fluorescence intensity at the end of each reaction. The same can be said in the case when the high concentration fibrils (1.0 mM, further on referred as HCF) are added to 1.0 mM insulin solutions. However, when the LCF are added to 1.0 mM insulin solutions, high seed concentrations yield a low fluorescence intensity, which then rises with decreased amount of seeds, eventually resulting in an intensity comparable to HCF. The opposite is observed when HCF are added to 0.2 mM insulin solutions, where

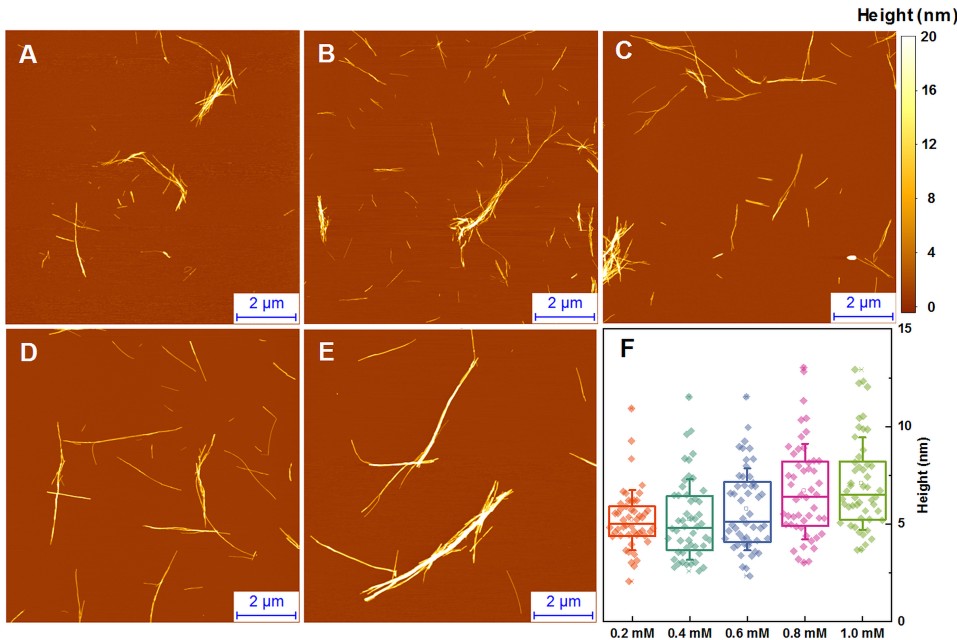

**Figure 2** **AFM analysis of insulin fibrils.** Insulin fibrils formed at 0.2 mM (A), 0.4 mM (B), 0.6 mM (C), 0.8 mM (D) and 1.0 mM (E) concentrations. Insulin fibril height distribution with box plots indicating the interquartile range and errors bars are for one standard deviation (F). Sample size for each ANOVA test was 50.

high initial fibril concentrations yield an intensity comparable to the seed conformation (when accounted for fibril concentration at the end of the reaction) and an intensity similar to LCF when the seed concentration becomes minimal.

In order to further confirm the self-replication ability of both conformations, fibrils formed during seeded aggregation were examined by FTIR and their spectra were compared to the unseeded aggregation fibril spectra (Figs. 4A–4D). The results show that when a large concentration of preformed fibrils is added to either 0.2 mM or 1.0 mM insulin solutions, the seed self-replicates and maintains its initial secondary structure. On the other hand, when a low concentration of seed is added, the resulting FTIR spectra are similar to their respective environment conformations, rather than the seed.

### Seeded fibril morphology

When large amounts of sonicated aggregates are used, there is minimal difference in the length and distribution of fibrils (Figs. 5A–5D), likely due to the large amount of aggregation centers. When the amount of seed used is low, the fibril length and distribution is similar to unseeded aggregation (Figs. 5E–5H). Fibril height distribution reveals a similarity between almost all conditions, except for when 1.0 mM insulin is seeded with low concentrations of either conformation (Fig. 5I), where the height distribution is comparable to unseeded nucleation. Additional AFM images of these conditions are available as supplementary information (Fig. S3).

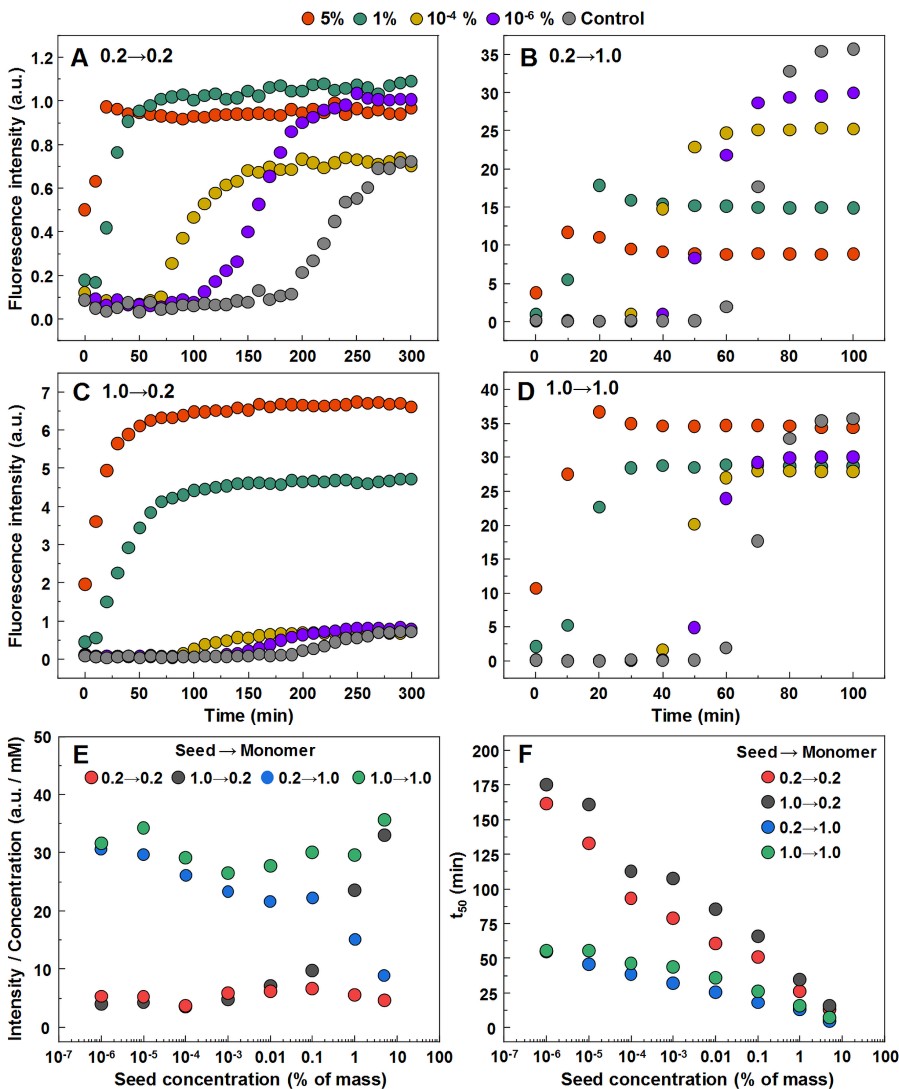

**Figure 3 Seeded aggregation of insulin with a range of preformed fibrils.** Aggregation kinetics of insulin where the LCF are added to 0.2 mM insulin solutions (A), LCF to 1.0 mM (B), HCF to 0.2 mM (C) and HCF to 1.0 mM (D). Aggregation half-time ($t_{50}$) dependence on concentration and type of seed added (E). ThT fluorescence intensity and fibril concentration ratio dependence on added seed concentration (F). Each data point is the average of four repeats.

## DISCUSSION

The first and most apparent difference between the samples, aggregated at different protein concentrations is their ability to enhance ThT fluorescence. A very similar effect was reported in case of protein-concentration-dependent polymorphism of glucagon amyloid fibrils (*Pedersen et al., 2006*). A 10-fold increase in ThT binding positions is highly improbable and slightly different fibril size distribution seen in AFM images could not strongly affect the number of binding positions, so a more appropriate explanation could be changes in the fibril's surface, facilitating a different ThT binding mode, as insulin

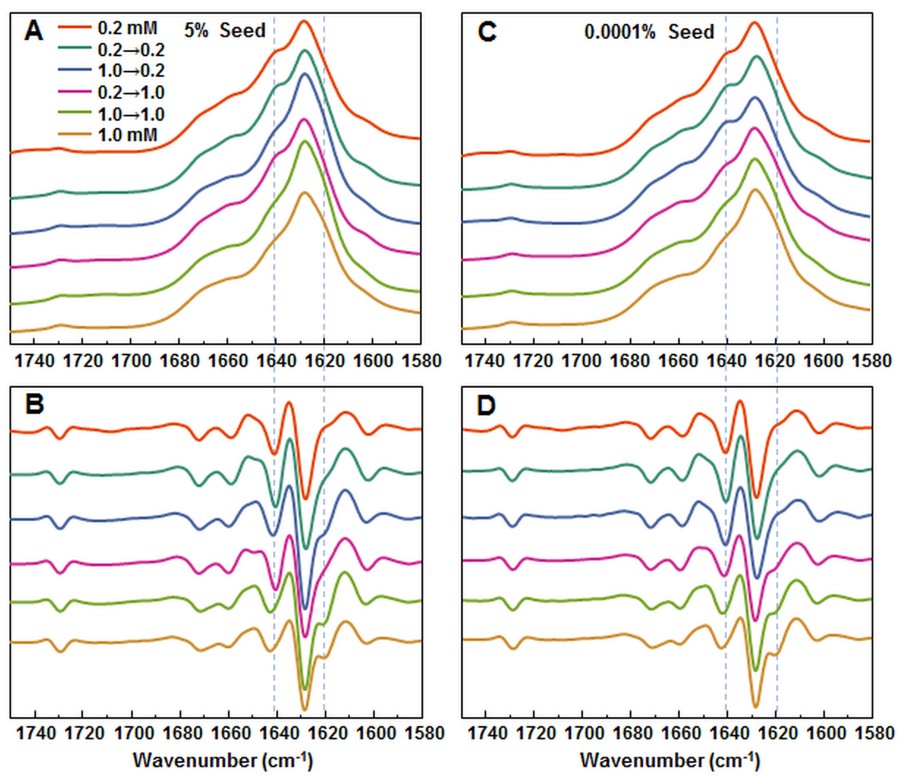

**Figure 4** **FTIR analysis of seeded aggregates.** Absorption and second derivate spectra of insulin fibrils when 5% (A and B, respectively) and 0.0001% (C and D, respectively) preformed fibrils are added.

fibrils have been shown to possess more than one way of incorporating ThT molecules (*Groenning et al., 2007*). Similar differences in ThT fluorescence were observed with different conformations of alpha-synuclein fibrils and attributed to the different binding of ThT molecules (*Sidhu et al., 2018*), so we can hypothesize that a low protein concentration leads to a different conformation of insulin amyloid fibrils.

Protein-concentration-dependent polymorphism of insulin amyloid fibrils is supported also by different FTIR spectra. In fact, spectral differences are rather minor in comparison to the ones observed between spectra of previously reported insulin conformations (*Dzwolak et al., 2004*; *Sneideris et al., 2015*), but the hallmark of each spectrum is conserved in seeding experiments (Figs. 4A and 4B), which supports the hypothesis of different amyloid conformations. Comparison of FTIR spectra to the previously reported data (*Sneideris et al., 2015*) suggests that the fibril conformation formed at higher insulin concentrations is the same as previously reported, while the one formed at lower concentrations falls out of the previously proposed scheme (*Sneideris et al., 2015*).

Atomic force microscopy data does not add much of information. It seems that average size of spontaneously formed fibrils slightly increases with higher protein concentration (Fig. 2), but this effect does not depend on the type of seeds (Fig. 5).

Currently we are aware of two mechanisms of seed-induced aggregation. One is amyloid fibril elongation via attachment and refolding of protein molecules at seed fibril ends,

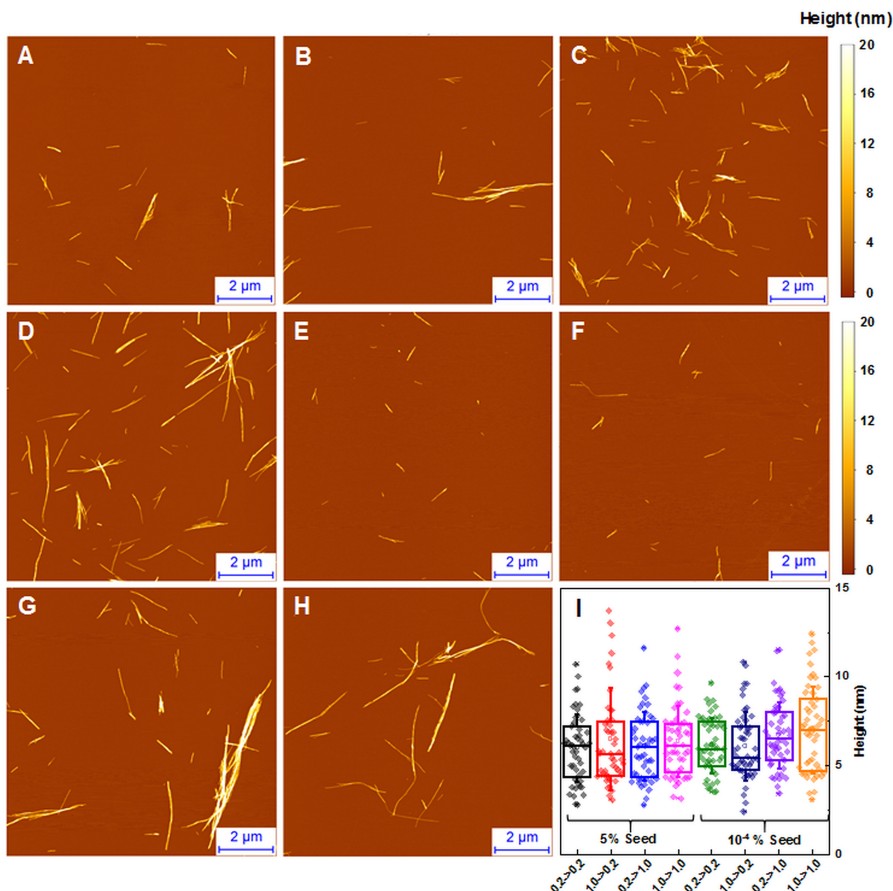

**Figure 5  AFM analysis of seeded insulin fibrils.** Insulin fibrils resulting from seeding 0.2 mM insulin with LCF (A, E), 0.2 mM with HCF (B, F), 1.0 mM with LCF (C, G), 1.0 mM with HCF (D, H) using 5% or $10^{-4}$ % of preformed fibrils respectively. Insulin fibril height distribution with box plots indicating the interquartile range and errors bars are for one standard deviation. Sample size for each ANOVA test was 50.

another one is formation of new aggregation nuclei catalyzed by the surface of seeds (often referred as secondary nucleation (*Meisl et al., 2016*; *Foderà et al., 2008*; *Scheidt et al., 2019*; *Törnquist et al., 2018*; *Bunce et al., 2019*)). Previously we have demonstrated that in case of cross-seeding of different environment-induced conformations of prion protein amyloid fibrils, the conformational template can self-propagate only via elongation mechanism, while surface induced nucleation only speeds up the aggregation process, but the conformation is defined by the environment conditions (*Sneideris, Milto & Smirnovas, 2015*). Recently similar observations were reported on amyloid beta (*Brännström et al., 2018*) and alpha synuclein (*Peduzzo, Linse & Buell, 2019*). Our cross-environment seeding data on insulin follows the same path. With higher amount of seeds, the aggregation kinetic curves are exponential, which means that the majority of the protein is aggregated via elongation of seeds—in such case the final relative ThT fluorescence intensity and FTIR spectra of seeds and final aggregates are very similar. Lowering the amount of seeds

leads to sigmoid kinetic curves which means that the majority of the protein is aggregated via new-formed nuclei and seeds are mainly employed as catalyzers—in such case the final relative ThT fluorescence intensity and FTIR spectra of seeds and final aggregates are different.

## CONCLUSIONS

Generally, in the seeded growth experiment of amyloid fibrils one expects self-replication of seed conformation. Here we showed that such expectations are valid only at certain circumstances—amyloid fibrils self-replicate their conformation only via elongation, or else the conformation of aggregates is environment-dependent. As similar conclusions were previously derived in studies of prion protein, amyloid beta, and alpha-synuclein, there may be enough data to consider it as a general way for self-replication of different amyloid fibril conformations.

## ACKNOWLEDGEMENTS

The authors acknowledge Prof. G. Niaura from the Center of Physical Sciences and Technology for the access to FTIR.

### Funding
The authors received no funding for this work.

### Competing Interests
The authors declare there are no competing interests.

### Author Contributions
- Andrius Sakalauskas and Mantas Ziaunys conceived and designed the experiments, performed the experiments, analyzed the data, prepared figures and/or tables, authored or reviewed drafts of the paper, approved the final draft.
- Vytautas Smirnovas conceived and designed the experiments, analyzed the data, contributed reagents/materials/analysis tools, authored or reviewed drafts of the paper, approved the final draft.

### Data Availability
The raw data are available as Supplementary Files.

### Supplemental Information
Supplemental information for this article can be found online at http://dx.doi.org/10.7717/peerj.8208#supplemental-information.

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
