# Peer review of "Concentration-dependent polymorphism of insulin amyloid fibrils"

_PeerJ, doi:10.7717/peerj.8208_

## Round 0.1 · original submission · Minor Revisions

As you see, all three reviewers are very positive. Reviewer3 has some suggestions of citations that you should add and reviewer 1 has a number of minor suggestions, which you might want to address in a revised manuscript, before it will be finally accepted.

·

Basic reporting

The article is clearly written, but in places features a somewhat clumsy language. For example I would advise to avoid phrases like "When the 0.2 mM-formed fibril conformation is added" and replace them by something like "When the LCF are added", whereby LCF stands for low concentration fibrils.
Also, the language should be proof read for general grammar, mainly missing articles. Example: “Ability of the same protein to adopt distinct pathogenic conformations” should read “The ability of the same protein to adopt distinct pathogenic conformations”

The structure of the article is clear, there is no superfluous material, and the literature citations are mostly adequate. I would recommend adding one or two references that discuss the existence of secondary nucleation pathways in the case of insulin aggregation, e.g. the work by Fodera et al., J. Phys. Chem. B 2008.

Raw data is provided in an easily accessible form.

Experimental design

The experiments are well described and designed. The research question is also well-defined and a very important one. Secondary nucleation of amyloid fibrils is receiving increased attention in recent years, as the potentially dominant route towards the generation of new aggregates. One of the key questions is whether secondary nucleation is able to propagate structural characteristics of the seed fibrils, as well as in the case of fibril growth, where this has already been demonstrated in multiple studies. The present study addresses this question in a clever way, by exploiting the concentration-dependent polymorphism of insulin aggregation. Despite the fact that the amyloid fibril formation experiments in this study are performed under highly non-physiological conditions, the study is highly relevant for the field of amyloid fibril formation and the link with disease. This is because very little is known about the molecular mechanisms and determinants of fibril surface-catalysed secondary nucleation, and therefore any mechanistic insight gained for any amyloid system is a valuable contribution at the current state of knowledge.
The conclusions of the study are based on the results of three types of experimental characterisations of fibrils: ThT fluorescence intensity (normalised by fibril mass), FT-IR spectroscopic signatures of fibrils and AFM analysis. All three techniques lead to the same conclusions, which makes the results very robust.

Validity of the findings

Overall, the study leaves no significant doubts as to the validity of the findings. I just have a few minor suggestions that might improve it further:

1) It would help to have also the unseeded data plotted in Figure 3, in order to help visualise the degree of seeding achieved by the lowest employed seed concentrations.

2) For the FT-IR and ThT experiments, it support the conclusions if the authors measured the degree of completion of the aggregation reaction at the plateau, at least in some select cases. With that I mean that the authors should centrifuge the samples and quantify the concentration of soluble insulin in the supernatant. This is in principle necessary before the authors can normalise the ThT signal by fibril mass. It needs to be shown that the sample converts (near) quantitatively into fibrils.

3) It would also help to show an FT-IR spectrum of monomer, such that it can be established that the differences in the observed FT-IR spectra between high and low concentration fibrils do not stem from different monomer contributions.

Reviewer 2 ·

Basic reporting

No comment.

Experimental design

No comment

Validity of the findings

No comment

Additional comments

The manuscript is clearly written and the authors findings are well explained

Reviewer 3 ·

Basic reporting

Some references should be cited:

On line 32-33: Doig AJ et al. ACS Chem Neurosci. 2017:1435-1437; Nasica-Labouze J et al. Chem Rev. 2015 May 13;115(9):3518-63; Man VH et al. J Phys Chem B. 2017 Jun 22;121(24):5977-5987.

On line 37: Nguyen PH et al. J Phys Chem B. 2014 Jan 16;118(2):501-10; Pellarin R et al. J Am Chem Soc. 2010 Oct 27;132(42):14960-70.

Later in the discussion: Scheidt T et al. Sci Adv. 2019 Apr 17;5(4):eaau3112; Törnquist M,
et al. Chem Commun (Camb). 2018 Aug 2;54(63):8667-8684; Bunce SJ et al.
Sci Adv. 2019 Jun 21;5(6):eaav8216.

Experimental design

well done

Validity of the findings

interesting and novel results are well identified.

Additional comments

The authors report a nice experimental study on the aggregation kinetics of insulin.
My only comments is to cite the references of litterature that I have reported.

---

## Round 0.2 · accepted · Accept

Your manuscript has been appropriately revised following the reviewers suggestions.